# Activity Patterns, Population Dynamics, and Spatial Distribution of the Stick Tea Thrips, *Dendrothrips minowai*, in Tea Plantations

**DOI:** 10.3390/insects14020152

**Published:** 2023-02-01

**Authors:** Fengge Zhang, Xiaoming Cai, Limeng Jin, Guojun Yang, Zongxiu Luo, Lei Bian, Zhaoqun Li, Nanxia Fu, Zongmao Chen, Guochang Wang, Chunli Xiu

**Affiliations:** 1Tea Research Institute, Chinese Academy of Agricultural Sciences, Hangzhou 310008, China; 2School of Resources and Environment, Henan Institute of Science and Technology, Xinxiang 453003, China; 3Key Laboratory of Biology, Genetics and Breeding of Special Economic Animals and Plants, Ministry of Agriculture and Rural Affairs, Hangzhou 310008, China; 4Shaoxing Royal Tea Village Co., Ltd., Shaoxing 312000, China; 5Hangzhou Fuhaitang Tea Ecological Technology Co., Ltd., Hangzhou 310024, China

**Keywords:** *Dendrothrips minowai*, activity height, diurnal activity pattern, spatial distribution, population dynamics

## Abstract

**Simple Summary:**

We studied the activity patterns, population dynamics, and spatial distribution of *Dendrothrips minowai* Priesner, one of the most destructive pests of tea plants in tea plantations. A large proportion of *D. minowai* individuals were caught in traps placed at heights ranging from 5 cm below to 25 cm above the position of tender leaves at the top of the tea plant, with most captures at a height of 10 cm above this position. The flight activity of *D. minowai* was highest from 10:00 to 16:00 h on sunny days in the spring and from 06:00 to 10:00 h and from 16:00 to 20:00 h in the summer. The distributions of *D. minowai* females and nymphs on leaves were aggregated according to Taylor’s power law and Lloyd’s patchiness index. The *D. minowai* populations were dominated by females, and the density of males was high in June. The seasonal prevalence of tea thrips captured with sticky traps in the field was bimodal; adult thrips overwintered on the bottom leaves. The peak periods of activity were from April to June and from August to October. This work provides new insights that have implications for enhancing the efficacy of measures to control *D. minowai*.

**Abstract:**

The stick tea thrips, *D. minowai* Priesner (Thysanoptera: Thripidae), is one of the most economically significant thrips pests of tea (*Camellia sinensis* (L.) O. Ktze.) in China. Here, we sampled *D. minowai* in tea plantations from 2019 to 2022 to characterize its activity patterns, population dynamics, and spatial distribution. A large proportion of *D. minowai* individuals were caught in traps placed at heights ranging from 5 cm below to 25 cm above the position of tender leaves at the top of the tea plant, and the greatest number of individuals were captured at a height of 10 cm from the position of tender leaves at the top of the tea plant. Thrips were most abundant from 10:00 to 16:00 h in the spring and from 06:00 to 10:00 h and from 16:00 to 20:00 h on sunny days in the summer. The spatial distribution of *D. minowai* females and nymphs was aggregated on leaves according to Taylor’s power law (females: *R*^2^ = 0.92, *b* = 1.69 > 1; nymphs: *R*^2^ = 0.91, *b* = 2.29 > 1) and Lloyd’s patchiness index (females and nymphs: *C* > 1, *Ca* > 0, *I* > 0, *M**/*m* > 1). The *D. minowai* population was dominated by females, and male density increased in June. Adult thrips overwintered on the bottom leaves, and they were most abundant from April to June and from August to October. Our findings will aid efforts to control *D. minowai* populations.

## 1. Introduction

Thrips (Insecta: Thysanoptera) are major pests of agricultural and horticultural crops around the world [1,2,3]. Many thrips are pests of commercial crops due to the direct damage they cause by feeding on developing flowers, fruits, vegetables, or leaves, affecting yield or cosmetic appearance [4,5]. Thrips may also serve as vectors for plant diseases, such as tospoviruses [6,7]. In most annual vegetable and row crop production systems, along with climatic variables, the seasonal availability of host plants proudly impacts thrips population dynamics [8,9]. However, in perennial plantation crops where the host is available throughout the year, thrips population dynamics and their dispersal patterns are largely influenced by climatic variables [4]. Therefore, in perennial agroecosystems, understanding yearly thrips population dynamics in the field and their dispersal patterns on host plants are very important in developing effective integrated pest management strategies [10]. The information can be used to predict thrips abundance on host plants in the field, and this may suggest further ways of developing their potential for pest management.

Thrips have become a threatening pest to tea plants (*Camilla sinensis* (L.) O. Ktze.) in China, which is one of the most important tea-producing and tea-exporting countries in the world [11,12]. A total of 28 thrips species have been documented in tea plantations in China, including *D. minowai* Priesner, *Scirtothrips dorsalis* Hood, and *Mycterothrips gongshanensi* Li, Li, and Zhang [13,14,15]. The stick tea thrips, *D*. *minowai* has become an increasingly significant pest of tea plants in China in recent years [16,17]. *D. minowai* induces direct damage to tea plants by sucking nutrients from the leaflets; whether it can also be a vector of viruses remains unknown. The presence of stripes and scarring along the leaf veins and blades on the abaxial and adaxial leaf surfaces is a sign of feeding damage, and heavy *D. minowai* infestations can lead to the gradual loss of leaf color, leaf stiffness, and decreases in tea yield and quality [18,19].

Insecticides are often used to control *D. minowai* in conventional tea plantations, and the use of insecticides can have negative environmental effects, decrease the abundance of beneficial natural enemies, and favor the evolution of resistance in thrips populations [20,21,22]. Thus, the frequency and timing of insecticide applications are critically important for the sustainable management of *D. minowai* populations. Knowledge of the abundance of thrips on tea plants is important for understanding seasonal variation in the activity of thrips, including the timing of infestations [23]. Knowledge of the spatial distribution of thrips on tea plants is also important for the development of strategies to control their populations [24]. The aims of this study were to monitor the flight heights of thrips, characterize their daily activity patterns, clarify the spatial distribution and population dynamics of *D. minowai* in tea fields, and provide data that will aid integrated thrips management programs in China.

## 2. Materials and Methods

### 2.1. Study Areas

We made observations of the flight height, diurnal activity patterns, spatial distribution pattern, sex ratio, and seasonal abundance of *D. minowai* in insecticide-free tea plantations (tea cultivar: Longjing 43) of Shaoxing Royal Tea Village Co., Ltd., Zhejiang Province, China (29.94° N, 120.71° E) in the District of Yuecheng, Shaoxing City, Zhejiang Province, China. We also studied the spatial distribution and sex ratio of this thrips species in the above tea plantation and three other tea plantations: Hangzhou Fuhaitang Tea Ecological Technology Co., Ltd. (tea cultivar: Longjing population) in the District of Xihu, Hangzhou City, Zhejiang Province (30.13° N, 120.03° E); Zhejiang Camel Jiuyu Organic Food Co., Ltd. (tea cultivar: Jiukeng) in the District of Yuhang, Hangzhou City (30.40° N, 119.90° E); and Shengzhou Tea Comprehensive Experimental Base, Tea Research Institute, Chinese Academy of Agricultural Sciences in Shaoxing City (tea cultivar: Zhongcha 108) (29.75° N, 120.83° E) (Figure 1).

### 2.2. Materials and Thrips Identification

Blue sticky traps made from PVC (10 × 25 cm) purchased from Hangzhou Yihao Agricultural Technology Co., Ltd., Zhejiang Province, China, were used to characterize the flight height and diurnal activity patterns of thrips. Actually, we identified *D. minowai* based on their morphological characteristics reported in the literature.

### 2.3. Flight Height Observations

The flight heights of *D. minowai* adults were evaluated using commercially available colored sticky traps. Blue sticky traps were used because they have been shown to be effective in attracting various other thrips species [25,26]. Traps were hung on branches at various heights below (negative numbers) and above (positive numbers) the position of tender tea leaves at the top of the tea plant (−35, −30, −25, −20, −15, −10, −5, 0, 5, 10, 15, 20, 25, 30, and 35 cm from the position of tender tea leaves at the top of the tea plant canopy; Figure 2a), and each trap was separated by a distance of at least 5 m. Five traps at each height (treatments) were placed randomly throughout organic tea plantations of Shaoxing Royal Tea Village Co., Ltd., from 2 May to 4 May 2022. The total numbers of thrips on each trap were recorded after 2 d.

### 2.4. Diurnal Activity Patterns

Following previous studies, observations of the diurnal activity patterns of *D. minowai* were made on sunny, cloudy, and rainy days [27]. Thrips were not active at night; the number of thrips caught at night was counted once; however, the number of thrips captured during the day was counted every 2 h (specifically, counts were conducted at 6:00, 8:00, 10:00, 12:00, 14:00, 16:00, 18:00, and 20:00). Sampling was conducted at organic tea plantations of Shaoxing Royal Tea Village Co., Ltd., in the spring (24 April 2022: sunny day; 25 April 2022: rainy day; 26 April 2022: cloudy day) and the summer (24 June 2022: sunny day; 26 June 2022: cloudy day; 28 June 2022: rainy day). Blue sticky cards were placed 10 cm away from the surface of tea leaves, and there was a distance of 5 m between each trap to minimize interference between traps. Five traps were randomly placed at the study sites during each sampling period. The number of thrips caught per sticky trap was recorded.

### 2.5. Spatial Distribution and Sex Ratio

The spatial distribution and sex ratio of *D. minowai* on leaves were studied in tea plantations at four sites (Hangzhou Fuhaitang Tea Ecological Technology Co., Ltd.; Zhejiang Camel Jiuyu Organic Food Co., Ltd.; Shaoxing Royal Tea Village Co., Ltd.; and Shengzhou Tea Comprehensive Experimental Base) in Zhejiang Province, China. To characterize the spatial distribution of *D. minowai*, we visually inspected tea plants for morphological indicators of female and nymph *D. minowai* presence [28]. Our previous observations suggest that male *D. minowai* adults are rarely found on tea leaves and spend most of their time hiding in tea bushes. This suggests that the above sampling method of visually inspecting plants is not effective for detecting *D. minowai* males. Therefore, to evaluate the sex ratio, we visually inspected plants for the presence of female and male *D. minowai* adults using knockdown techniques [29], which involved holding tea branches over a rectangular 40 × 20 × 10 cm white-colored pan and striking the branch five times; the numbers of females and males that fell into the pan were then counted [28].

### 2.6. Seasonal Abundance

The seasonal abundance of *D. minowai* on tea leaves was monitored at weekly intervals between April 2019 and October 2022 in organic tea plantations of Shaoxing Royal Tea Village Co., Ltd. To measure thrips abundances, we divided the study area into plots of 20 × 30 m. We then sampled *D. minowai* adults from the 100 tea leaves at five random points within each plot. The numbers of *D. minowai* adults on the upper (second leaf under the tender shoot), middle, and bottom leaves were estimated using the method described above.

### 2.7. Statistical Analyses

All data were checked for normality and equality of variances prior to statistical analysis. Datasets that did not meet assumptions were square-root transformed to meet the requirements of equal variances and normality. Differences in the numbers of thrips per trap, at different heights, and during different periods were determined using analysis of variance (Minitab 13, Minitab Inc., State College, PA, USA).

The means (*m*) and variance (*V*) of the densities of thrips were calculated. Means and variances of *D. minowai* were modeled according to Taylor’s power law (TPL) [lg(*V*) = lg*a* + *b*lg(*m*)], where *a* is a sampling factor, and *b* is the aggregation parameter. The distribution is considered regular if *b* < 1; random if *b* = 1; and aggregated if *b* > 1 [30]. The spatial distribution of the thrips was analyzed using density data and Lloyd’s patchiness index [31,32]. Parameters were obtained using the following model: diffusion coefficient *C* = *V*/*m*, diffusion index *Ca* = (*V* − *m*)/*m*^2^, negative binomial distribution *K* = *m*/(*V*/*m* − 1), index of clumping *I* = *V*/*m* − 1, mean crowding intensity *M** = *m* + *V*/*m* − 1, and aggregation index *M**/*m*. *C <* 1, *I* < 0, *Ca <* 0, *M**/*m* < 1 represents a regular distribution, *C* = 1, *I* = 0, *Ca* = 0, *M**/*m* = 1 represents a random distribution, and *C* > 1, *I >* 0, *Ca* > 0, *M**/*m* > 1 represents an aggregated distribution.

Graphs of the flight height, daily flight activity, and seasonal distribution of *D. minowai* were made using GraphPad Prism 7.0, and graphs of the linear relationships between variances and means of *D. minowai* were made using OriginPro 2021. All analyses were conducted using SAS 9.4.

## 3. Results

### 3.1. Flight Height

The numbers of *D. minowai* captured on blue sticky traps varied at different heights (*F*_14,60_ = 385.65, *p* < 0.001) (Figure 2b). Overall, traps ranging from 5 cm below to 25 cm above the position of tender leaves at the top of the tea plant had a high number of thrips. Most thrips were captured at a height of 10 cm above the position of tender leaves at the top of the tea plant (Figure 2b).

### 3.2. Diurnal Activity Patterns

The daily flight activity of the stick tea thrips *D. minowai* was examined using blue sticky traps in tea plantations. We found that thrips flight activities were affected by both weather and temperature (Figure 3). In the spring, thrips were most abundant between 10:00 and 16:00 h on sunny days, and their abundances declined from 16:00 to 20:00; they were largely inactive from 20:00 to 08:00 h. The number of thrips captured in traps was significantly lower on rainy or cloudy days than on sunny days between 10:00 and 16:00 h; the daily flight curve was unimodal (Figure 3a).

In the summer, the daily flight curve was bimodal on hot, sunny days (Figure 3b). Specifically, thrips were more abundant between 06:00 and 10:00 h and between 16:00 and 20:00 h; only a few thrips were found between 12:00 and 16:00 h and between 20:00 and 06:00 h. On cloudy days, thrips were captured on sticky traps during the entire sampling period from 06:00 to 20:00 h; a bimodal daily flight curve was also observed on these days. Few thrips were captured in traps throughout the sampling period on rainy days.

### 3.3. Spatial Distribution and Sex Ratio

Variances and means were significantly related according to Taylor’s power law (females: *R*^2^ = 0.92, *p* < 0.0001, *b* = 1.69 > 1; nymphs: *R*^2^ = 0.91, *p* < 0.0001, *b* = 2.29 > 1), indicating that the distribution of *D. minowai* females and nymphs in the four different tea plantations was aggregated (Figure 4a,b). Lloyd’s patchiness index indicated that the distribution of *D. minowai* females on tea leaves in Yuecheng District, Shaoxing, China, was aggregated from April to June 2021 and, in Xihu District and Yuhang District, Hangzhou and in Shengzhou County, Shaoxing, from April to June in 2022 (*C* > 1, *Ca* > 0, *I* > 0, *M**/*m* > 1) (Table 1). Lloyd’s patchiness index indicated that the distribution of *D. minowai* nymphs on tea leaves was aggregated in Yuecheng District, Shaoxing, from April to June 2022 (*C* > 1, *Ca* > 0, *I* > 0, *M **/*m* > 1) (Table 2). In general, the spatial distributions of *D. minowai* females and nymphs were relatively stable within different fields and different periods, respectively.

The proportions of female and male *D. minowai* adults in different periods at different sites varied (Table 1). The proportions of female thrips in tea plantations were higher than those of males throughout most of the sampling period, indicating that *D. minowai* populations were dominated by females in the tea plantations, especially as the density of the thrips population increased (from April to the end of May). However, the density of *D. minowai* males increased in early June and eventually outnumbered females by late June.

### 3.4. Seasonal Abundance of D. minowai

Annual cycles were observed in the *D. minowai* female population in tea plantations, with a bimodal type of occurrence, the two highest densities occurring in April to June and August to October on tea leaves, regardless of the cultivars and years in Zhejiang Province, China (Figure 5). However, in 2022, the number of *D. minowai* females was near zero, different from the months of August to October 2019–2021. The abnormal phenomenon was caused by the continuous extremely high temperature and drought from June to August 2022 (Appendix A).

Adult thrips overwintered on the bottoms of leaves. *D. minowai* adults began to move to the lower middle leaves in tea plantations in October and remained on the bottom leaves until the following March.

## 4. Discussion

Characterizing the flight heights and diurnal activity patterns of thrips is important for accurately estimating their densities and dispersal patterns, as well as developing pesticide application strategies. The flight activity patterns of insects are related to their responses to sunlight, temperature, and relative humidity [24,31,33]. In our study, the flight activity patterns of *D. minowai* on sunny days differed in the spring and summer. *D. minowai* flight activity peaked from 10:00 to 16:00 h in the spring; however, the peaks of their flight activity were from 06:00 to 10:00 h and from 16:00 to 20:00 h in the summer (Figure 3). They appear to avoid flying in temperatures below 20 °C or above 30 °C (Appendix A). No flight activity was observed at night (Figure 3). *Frankliniella occidentalis* Pergande females are immobile at midday and at night [34], and *Thrips imaginis* Bagnall, *T. hawaiiensis* Morgan, *S. dorsalis*, *Megalurothrips usitatus* Bagnall, and *F. schultzei* Trybom seek refuge on their host plants during the hottest times of day, which corresponds to the period when their densities are highest [35,36]. The number of thrips captured on traps was significantly lower on rainy days than on sunny days. The decreased abundance of thrips on rainy days might stem from the effect of temperature, solar radiation, or an effect of humidity.

In our study, the distribution of *D. minowai* females from April to June at all sites was aggregated according to Taylor’s power law (*b* > 1) and Lloyd’s patchiness index (*C* > 1, *Ca* > 0, *I* > 0, *M**/*m* > 1) (Table 1). These results indicate that the aggregated distribution of females was not affected by tea variety and geographic region. The distributions of other thrips species have also been shown to be aggregated. For example, the distribution of *F. occidentalis* was significantly aggregated on cucumber, cotton, tomato, and strawberry [37,38,39,40], the distribution of *F. schultzei* was significantly aggregated on cucumber [41], the distribution of *Pezothrips kellyanus* Bagnall was significantly aggregated in citrus groves [42], and the distribution of *S. dorsalis* was significantly aggregated on chili plants [43]. The distribution of nymphs was more aggregated than that of females (nymphs: *b* = 2.29 > females: *b* = 1.69) in tea plantations (Figure 4). This pattern has been observed in many other thrips species; nymphs aggregate during the early nymphal stages mainly because of their limited mobility, and they become less aggregated as their mobility increases [44]. These findings are consistent with the results of previous studies showing that the distribution of nymphs is more aggregated than that of females in *F. occidentalis* on tomato flowers and on greenhouse cucumber leaves, as well as in *T. hawaiiensis*, *T. palmi* Karny, and *S. dorsalis* on their host plants [24,42,45].

Our study of the population dynamics of thrips across four years revealed two key periods in which the abundance of *D. minowai* and, thus, the damage that they induced to tea plants were highest in Zhejiang Province (Figure 5). This information can aid the management and control of thrips on tea leaves in different seasons. In addition, *D. minowai* adults overwintered on the bottom leaves. The full-bloom stage of tea plants runs from mid-October to late November, and most tea flowers are present on the lower to middle parts of tea plants [46]. Some *D. minowai* adults began to colonize the lower middle leaves in tea plantations starting in October. *D. minowai* adults overwinter on the bottom leaves from November until the following March. During this stage, volatiles such as beta-ocimene, farnesene, and methyl benzoate have been identified [47]. According to our previous research, *D. minowai* is attracted to the above three volatiles [17]. Thus, the presence of overwintering adult thrips in the lower middle part of the tea plants might stem from their attraction to the volatiles of tea flowers. However, more laboratory studies and fieldwork are needed to clarify the overwintering mechanism of *D. minowai*. In any case, the bottom tea leaves merit increased attention, given that many of them serve as overwintering sites for these thrips.

## 5. Conclusions

Our data on the flight heights and activity patterns of thrips indicate that blue sticky traps hanging at 10 cm above the position of tender leaves at the top of the tea plant were more effective on sunny days. The distribution of *D. minowai* females and nymphs was aggregated, and a bimodal type of occurrence was observed in the female population in tea plantations. Adult thrips overwinter on the bottom leaves. Thus, the application of pesticides on old bottom leaves during the winter months could reduce population densities and pesticide residues in the following year. The results of this study provide new insights that will aid the management of *D. minowai* populations in tea fields, as well as the development of integrated pest management programs to control *D. minowai* infestations.

## Figures and Tables

**Figure 1 insects-14-00152-f001:**
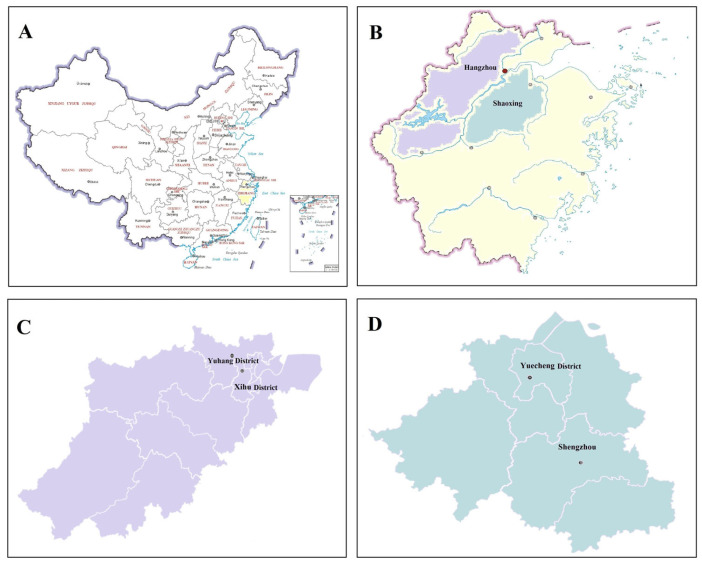
Map of China with the study areas shaded. Thrips trapping sites and study locations are indicated by the circles. (**A**). China; (**B**). Zhejiang Province; (**C**): Hangzhou City; (**D**): Shaoxing City.

**Figure 2 insects-14-00152-f002:**
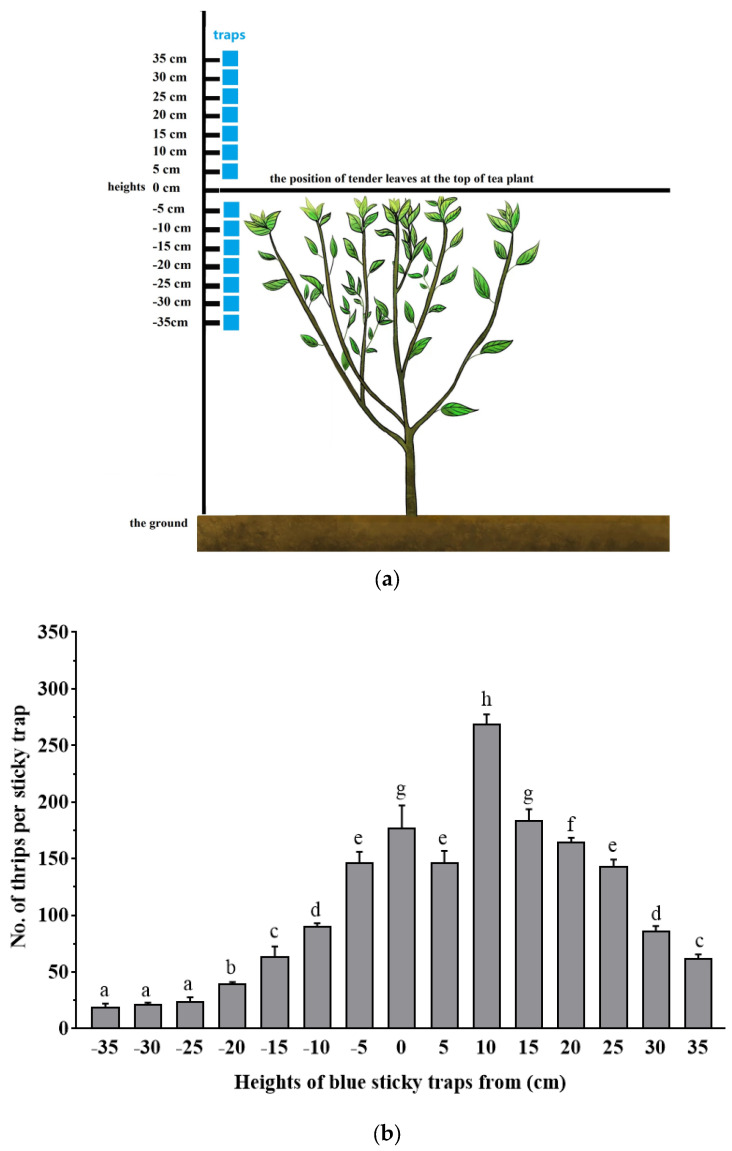
Schematic diagram of how the traps were placed at different heights from below (negative numbers) to above (positive numbers) the position of tender leaves at the top of the tea plant in tea plantations (**a**) and the numbers (mean ± SEM) of *D. minowai* adults captured in traps (**b**). Different lowercase letter denotes a difference at the *p* < 0.05 level, while the same lowercase letter was not significantly different (*p* > 0.05) (ANOVA followed by LSD).

**Figure 3 insects-14-00152-f003:**
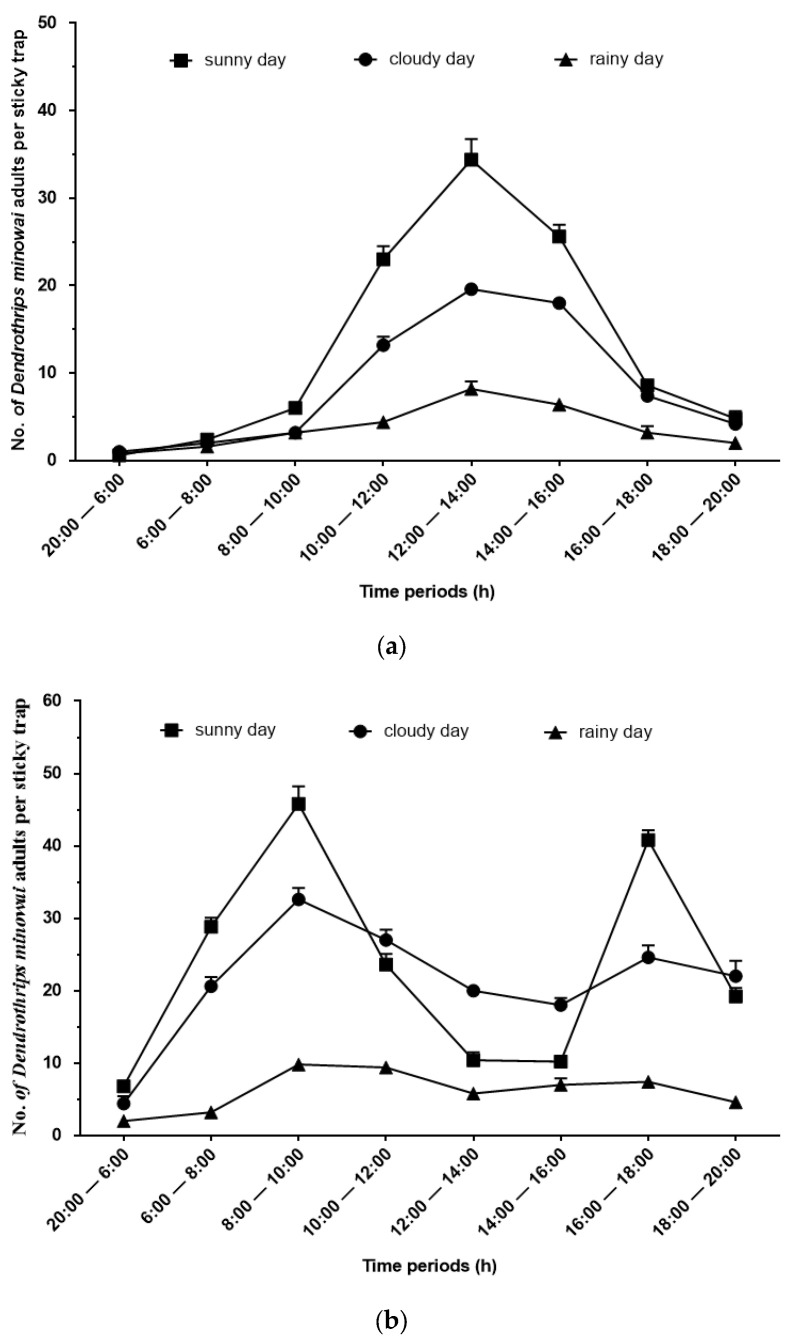
Numbers (mean ± SEM) of *D. minowai* adults captured in traps in different periods under different weather conditions. (**a**): Shaoxing in April 2022; (**b**): Shaoxing in June 2022. Five blue sticky traps were placed in the sampling area during each sampling period.

**Figure 4 insects-14-00152-f004:**
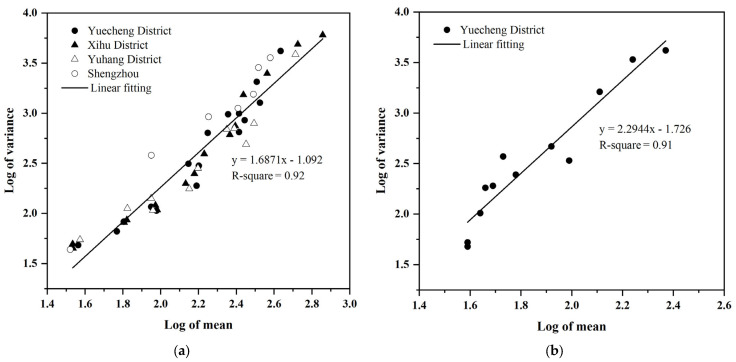
Linear relationship between variances and means of *D. minowai* on tea leaves in Zhejiang Province, China. (**a**): Females in different periods and four different sites from 2021 to 2022; (**b**): nymphs in Yuecheng District, Shaoxing, China, from April to June 2022.

**Figure 5 insects-14-00152-f005:**
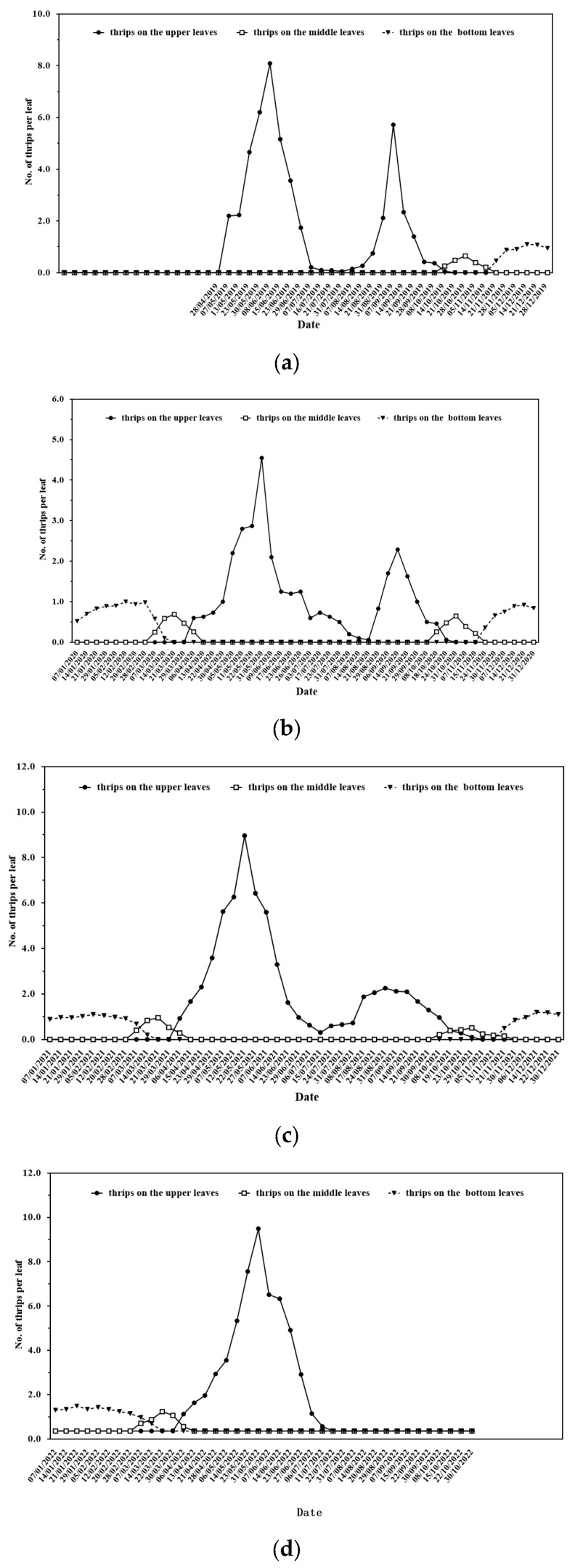
Abundance of *D. minowai* adults on different parts of tea leaves (upper leaves, middle leaves, and bottom leaves) in 2019–2022. (**a**) Shaoxing in 2019; (**b**) Shaoxing in 2020; (**c**) Shaoxing in 2021; and (**d**) Shaoxing in 2022.

**Table 1 insects-14-00152-t001:** The aggregation indexes of *D. minowai* females on leaves and the sex ratio (female:male) in different tea plantations in Zhejiang Province, China.

Sites		Date	Average Density (*m*)	Variance (*V*)	Diffusion Coefficient (*C*)	Diffusion Index (*Ca*)	Negative Binomial Distribution (*K*)	Index of Clumping (*I*)	Aggregating Index (*M**/*m*)	Sex Ratio (F:M)
Hangzhou	Xihu District	3 April 2022	34.4 ± 5.99	44.8	1.3023	0.0088	113.7846	0.3023	1.0088	3.59:1.00
		7 April 2022	64.0 ± 8.07	81.5	1.2734	0.0043	234.0571	0.2734	1.0043	3.74:1.00
		12 April 2022	93.8 ± 9.83	120.7	1.2868	0.0031	327.0796	0.2868	1.0031	3.09:1.00
		17 April 2022	150.8 ± 14.13	249.7	1.6558	0.0043	229.9357	0.6558	1.0043	2.99:1.00
		27 April 2022	232.2 ± 22.11	611.2	2.6322	0.0070	142.2608	1.6322	1.0070	2.12:1.00
		5 May 2022	274.0 ± 34.96	1527.5	5.5748	0.0167	59.8931	4.5748	1.0167	2.28:1.00
		16 May 2022	365.8 ± 44.66	2493.7	6.8171	0.0159	62.8834	5.8171	1.0159	1.65:1.00
		23 May 2022	531.2 ± 62.45	4875.2	9.1777	0.0154	64.9571	8.1777	1.0154	1.34:1.00
		29 May 2022	720.0 ± 69.56	6047.5	8.3993	0.0103	97.3064	7.3993	1.0103	1.21:1.00
		4 June 2022	248.2 ± 24.46	747.7	3.0125	0.0081	123.3298	2.0125	1.0081	1.00:1.38
		11 June 2022	170.0 ± 17.74	393.5	2.3147	0.0077	129.3065	1.3147	1.0077	1.00:2.10
		15 June 2022	135.6 ± 12.61	198.8	1.4661	0.0034	290.9392	0.4661	1.0034	1.00:2.36
		21 June 2022	95.8 ± 9.33	108.7	1.1347	0.0014	711.4450	0.1347	1.0014	1.00:1.51
		24 June 2022	66.2 ± 8.30	86.2	1.3021	0.0046	219.1220	1.3021	1.0046	1.29:1.00
		28 June 2022	34.2 ± 6.31	49.7	1.4532	0.0133	75.4606	1.4532	1.0133	1.41:1.00
	Yuhang District	4 April 2022	37.4 ± 6.62	54.8	1.4652	0.0124	80.3885	0.4652	1.0124	4.53:1.00
		14 April 2022	91.0 ± 9.27	107.5	1.1813	0.0020	501.8788	0.1813	1.0020	3.80:1.00
		25 April 2022	157.4 ± 15.04	282.8	1.7967	0.0051	197.5659	0.7967	1.0051	3.20:1.00
		15 May 2022	282.8 ± 19.81	490.7	1.7351	0.0026	384.6842	0.7351	1.0026	2.20:1.00
		24 May 2022	516.4 ± 55.62	3866.3	7.487	0.0126	79.6051	6.4870	1.0126	1.45:1.00
		28 May 2022	311.6 ± 25.16	791.3	2.5395	0.0049	202.4068	1.5395	1.0049	1.23:1.00
		2 June 2022	245.0 ± 23.77	706.5	2.8837	0.0077	130.0650	1.8837	1.0077	1.00:1.98
		9 June 2022	223.6 ± 23.55	693.3	301006	0.0094	106.4445	2.1006	1.0094	1.00:2.91
		16 June 2022	141.6 ± 11.88	176.3	1.2451	0.0017	577.8259	0.2451	1.0017	1.00:3.47
		22 June 2022	89.0 ± 10.64	141.5	1.5899	0.0066	150.8762	0.5899	1.0066	1.00:1.74
		27 June 2022	66.6 ± 9.48	112.3	1.6862	0.0103	97.0582	0.6862	1.0103	1.17:1.00
Shaoxing	Yuecheng District	1 April 2021	36.6 ± 6.22	48.3	1.3197	0.0087	114.4923	0.3197	1.0087	1.35:1.00
		6 April 2021	58.6 ± 7.28	66.3	1.1314	0.0022	445.9688	0.1314	1.0022	1.41:1.00
		10 April 2021	95.0 ± 9.23	106.5	1.1211	0.0013	784.7826	0.1211	1.0013	1.59:1.00
		15 April 2021	154.8 ± 12.30	189.2	1.2222	0.0014	696.6000	0.2222	1.0014	1.71:1.00
		23 April 2021	159.2 ± 15.47	299.2	1.8794	0.0055	181.0331	0.8794	1.0055	1.99:1.00
		29 April 2021	259.6 ± 22.79	649.3	2.5012	0.0058	172.9334	1.5012	1.0058	2.08:1.00
		4 May 2021	278.2 ± 26.10	851.7	3.0615	0.0074	134.9525	2.0615	1.0074	2.32:1.00
		12 May 2021	335.2 ± 31.91	1272.7	3.7968	0.0083	119.8496	2.7968	1.0083	2.99:1.00
		22 May 2021	431.2 ± 57.79	4175.2	9.6827	0.0201	49.6617	8.6827	1.0201	3.34:1.00
		27 May 2021	322.4 ± 40.56	2056.3	6.3781	0.0167	59.9468	5.3781	1.0167	3.23:1.00
		1 June 2021	259.8 ± 28.21	994.7	3.8287	0.0109	91.8438	2.8287	1.0109	2.24:1.00
		7 June 2021	227.4 ± 27.94	975.8	4.2911	0.0145	69.0951	3.2911	1.0145	1.80:1.00
		10 June 2021	177.4 ± 22.55	635.8	3.584	0.0146	68.6535	2.5840	1.0146	1.00:1.00
		14 June 2021	140.2 ± 15.84	313.7	2.2375	0.0088	113.2913	1.2375	1.0088	1.00:2.04
		23 June 2021	89.0 ± 9.65	116.5	1.309	0.0035	288.0364	0.3090	1.0035	1.00:2.26
		26 June 2021	63.8 ± 8.13	82.7	1.2962	0.0046	215.3672	0.2962	1.0046	1.00:3.37
		29 June 2021	33.2 ± 5.91	43.7	1.3163	0.0095	104.9752	0.3163	1.0095	2.00:1.00
	Shengzhou	8 May 2022	309.4 ± 35.19	1548.3	5.0042	0.0129	77.2688	4.0042	1.0129	3.71:1.00
		17 May 2022	329.2 ± 47.8	2855.7	8.6747	0.0233	42.8944	7.6747	1.0233	3.19:1.00
		25 May 2022	380.0 ± 53.51	3579	9.4184	0.0222	45.1391	8.4184	1.0222	2.58:1.00
		3 June 2022	256.4 ± 29.96	1121.8	4.3752	0.0132	75.9660	3.3752	1.0132	1.21:1.00
		11 June 2022	179.0 ± 27.17	922.5	5.1536	0.0232	43.0948	4.1536	1.0232	1.00:2.00
		18 June 2022	89.2 ± 17.45	380.7	4.2679	0.0366	27.2955	3.2679	1.0366	1.00:3.71
		25 June 2022	33.2 ± 5.91	43.7	1.3163	0.0095	104.9752	0.3163	1.0095	1.00:1.56

**Table 2 insects-14-00152-t002:** The aggregation indexes of *D. minowai* nymphs on leaves in tea plantations in Zhejiang Province, China.

Sites	Date	Average Density (*m*)	Variance (*V*)	Diffusion Coefficient (*C*)	Diffusion Index (*Ca*)	Negative Binomial Distribution (*K*)	Index of Clumping (*I*)	Aggregating Index (*M**/*m*)
Shaoxing (Yuecheng District)	10 April 2021	39.3 ± 6.46	52.2	1.3277	0.0083	120.0345	0.3277	1.0083
	15 April 2022	38.7 ± 6.18	47.8	1.2356	0.0061	164.0976	0.2356	1.0061
	23 April 2022	48.7 ± 12.40	192.2	3.9498	0.0606	16.4985	2.9498	1.0606
	29 April 2022	53.3 ± 17.30	373.9	7.0196	0.1130	8.8489	6.0196	1.1130
	4 May 2022	45.3 ± 12.04	181.1	3.9951	0.0661	15.1358	2.9951	1.0661
	12 May 2022	60.7 ± 14.05	246.7	4.0659	0.0505	19.7873	3.0659	1.0505
	18 May 2022	128.7 ± 36.00	1620.0	12.5907	0.0901	11.1009	11.5907	1.0901
	22 May 2022	172.0 ± 52.18	3403.3	19.7868	0.1092	9.1554	18.7868	1.1092
	27 May 2022	236.7 ± 57.46	4127.8	17.4413	0.0695	14.3946	16.4413	1.0695
	1 June 2022	96.8 ± 16.56	342.6	3.5391	0.0262	38.1231	2.5391	1.0262
	7 June 2022	84.0 ± 19.25	463.3	5.5159	0.0538	18.6011	4.5159	1.0538
	10 June 2022	44.0 ± 9.04	102.2	2.3232	0.0301	33.2519	1.3232	1.0301

## Data Availability

The data presented in this study are available on request from the corresponding authors.

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
