# Peer review of "Activity Patterns, Population Dynamics, and Spatial Distribution of the Stick Tea Thrips, *Dendrothrips minowai*, in Tea Plantations"

_insects, 2023, doi:10.3390/insects14020152_

Round 1

Reviewer 1 Report

 My manuscript by Zhang et al is a well-conducted field study on thrips population dynamics in organic tea plantations. In the manuscript, I could not find how authors identified the thrips up to the species level. Other than this, I have minor comments listed below. 

Line 21 and 36: how flight height is minus 5? Is it 5 to 25 cm? 

Line 301: Why humidity would not affect the thrips abundance? 

Line 341: correct spacing. 

What were the criteria used for the stick tea thrip identification? Write it somewhere in the material and method section.  

The introduction is very small. I know thrips are well-known insect pests around the world. But consider adding general information and general rationale behind the study in the introductory paragraphs. Then you can talk about thrips under tea cultivation as you have already done. I have suggested an addition below, change it as you like.  

Thrips (Insecta: Thysanoptera) are among the most important insect pests of vegetable, row, and perennial crops around the world (https://doi.org/10.1146/annurev.ento.51.110104.151044).  Several of the thrips species, because of their ability to transmit viruses are major insect pests in numerous crops ( https://doi.org/10.1093/jipm/pmab045; 10.1111/1744-7917.12721). Epub 2019 Oct 23.). In most annual vegetable and row crop production systems along with climatic variables, the seasonal availability of host plants proudly impacts thrips population dynamics. However, in perennial plantation crops where the host is available throughout the year, thrips population dynamics and their dispersal patterns are largely influenced by climatic variables (https://doi.org/10.1603/EN10066). Therefore, in perennial agroecosystems, understanding yearly thrips population dynamics in the field and their dispersal patterns on host plants are very important in developing effective integrated pest management strategies. The information can be used to predict thrips abundance in the field and on plants, thereby assisting in developing proactive management plans. 

 Thrips have become a threatening pest to tea plants (Camilla sinensis (L.) O. Ktze.) in China,……. 

Author Response

Response to Reviewer 1 Comments

Point 1: My manuscript by Zhang et al is a well-conducted field study on thrips population dynamics in organic tea plantations. In the manuscript, I could not find how authors identified the thrips up to the species level. Other than this, I have minor comments listed below.

Response 1: Thanks very much. Actually we identified the thrips based on morphological characteristics, we have been working on the researches of Dendrothrips minowai, such as the work of “Predatory functional evaluation of Orius sauteri against Dendrothrips minowai adults” has been accepted by “Journal of Plant Protection” of China, and the work of “Evaluation of Selected Plant Volatiles as Attractants for the Stick Tea Thrips Dendrothrips minowai in the Laboratory and Tea Plantation” has been accepted by “Insects”. Here in this manuscript, we forgot show readers the criteria used for the stick tea thrips identification, so we added the criteria in the section of “Materials and Methods”. Please see Line 108 to 112.

Point 2: Line 21 and 36: how flight height is minus 5? Is it 5 to 25 cm?

Response 2: Thanks very much. Our statement is not clear. So we revised these sentences as “A large proportion of D. minowai individuals were caught in traps placed at the heights ranged from minus 5 to 25 cm from the surface of tea leaves, with the heights of 10 cm captured the most”. Please see Line 21-23.

Point 3: Line 301: Why humidity would not affect the thrips abundance?

Response 3: Thanks very much. Our statement is not clear. The text has been revised accordingly. Please see Line 321.

Point 4: Line 341: correct spacing.

Response 4: Thanks very much. Spaces have been removed. Please see Line 362.

Point 5: What were the criteria used for the stick tea thrips identification? Write it somewhere in the material and method section.  

Response 5: Thanks very much. Here in this manuscript, we forgot show readers the criteria used for the stick tea thrips identification, so we added the criteria in the section of “Materials and Methods”. Please see Line 108 to 112.

Point 6: The introduction is very small. I know thrips are well-known insect pests around the world. But consider adding general information and general rationale behind the study in the introductory paragraphs. Then you can talk about thrips under tea cultivation as you have already done. I have suggested an addition below, change it as you like.  

Thrips (Insecta: Thysanoptera) are among the most important insect pests of vegetable, row, and perennial crops around the world (https://doi.org/10.1146/annurev.ento.51.110104.151044).  Several of the thrips species, because of their ability to transmit viruses are major insect pests in numerous crops ( https://doi.org/10.1093/jipm/pmab045; 10.1111/1744-7917.12721). Epub 2019 Oct 23.). In most annual vegetable and row crop production systems along with climatic variables, the seasonal availability of host plants proudly impacts thrips population dynamics. However, in perennial plantation crops where the host is available throughout the year, thrips population dynamics and their dispersal patterns are largely influenced by climatic variables (https://doi.org/10.1603/EN10066). Therefore, in perennial agroecosystems, understanding yearly thrips population dynamics in the field and their dispersal patterns on host plants are very important in developing effective integrated pest management strategies. The information can be used to predict thrips abundance in the field and on plants, thereby assisting in developing proactive management plans.

Thrips have become a threatening pest to tea plants (Camilla sinensis (L.) O. Ktze.) in China,…….

Response 6: Thanks for your wonderful suggestion. We have already adding general information and general rationale in the part of Introduction. Please see Line 51-65.

Reviewer 2 Report

All my remarks I have included in the manuscript

Author Response

Response to Reviewer 2 Comments

Point 1: Line 53-54: It is necessary to add the author name of the species name if you use it for the first time.

Response 1: Thanks for your advice. The text has been revised accordingly. Please see Line20, 33, 70 to 73.

Point 2: Line 81: is it the same cultivar as above? (Longjin 43), if yes - correct it.

Response 2: Thanks. These two cultivars are different, Longjin 43 and Longjin population are two cultivars.

Point 3: Line 91: Please, localize the study area (with dots or other symbols) on the map of China (A). It is not obvious to outsiders

Response 3: Thanks for your advice. It is difficult to localize the study area (with dots or other symbols) on the map of China (A), we localize Shaoxing City on picture B. Please see Line 1031.

Point 4: Line 92: Materials and Methods

Response 4: Thanks. The text has been revised as requested. Please see Line 89.

Point 5: Line 132-139: This method is incorrectly described. In the results in figure 5 (a-d), the dates are from January up to December according to the station and year.

Response 5: Thank you for this suggestion. Actually, our investigations are began from April 2019 to October 2022. Part of the month in the picture is absent, thus to keep the picture consistent, they were represented by a blank horizontal line. Please see Line 280 to 281of Fig. 5 (a-d).

Point 6: Line 154: ouhgth to be M*/m.

Response 6: Yes, it is. We have already revised this. Please see Line 186.

Point 7: Line 196: squares, dots and triangles ought to be a little bigger on both figures.

Response 7: Thank you for this suggestion. We have already adjusted dots and triangles larger on both figures. Please see Line 227.

Point 8: Line 238-240: The sex index in bisexual thrips populations is similar in most species - females live longer than males and appear earlier after overwintering, this is well documented in many studies. Sometimes males are ephemeric. It could be interesting to what was the sex index in the second summer generation but it was not researched.

Response 8: Thanks for interesting suggestion. 

Point 9: Line 295: It is necessary to cite the full name of the species if you use it for the first time.

Response 9: Thanks for your advice. We have already revised this. Please see Line 315 to 318, 332, 340 in the paper.

Point 10: Line 296: ?, cite the article: Insect Science (2010) 17, 535–541, DOI 10.1111/j.1744-7917.2010.01337.x   If you decide to cite it you have to check the numbering of articles [24]? - in Re

Response 10: Thanks. Yes, we cite it. It has now been revised. Please see Line 316.

Point 11: Line 298: [25-26] according to the References.

Response 11: Thank you for this suggestion. We have already adjusted the References. Please see the revised manuscript.

Point 12: Line 310, 319, 320.

Response 12: Thank you for this suggestion. Corrected them as abbreviations. Please see Line 330, 340-341.

Point 13: Line 336: Such a statement is not justified because adults overwintering on the lower leaves are often hidden and the use of pesticides is not effective in this case. I propose to delete the final sentence.

Response 13: Thank you for this suggestion. We have already deleted it. Please see Line 358-359.